# Real-Time Evaluation of Time-Domain Pulse Rate Variability Parameters in Different Postures and Breathing Patterns Using Wireless Photoplethysmography Sensor: Towards Remote Healthcare in Low-Resource Communities

**DOI:** 10.3390/s23094246

**Published:** 2023-04-24

**Authors:** Felipe Pineda-Alpizar, Sergio Arriola-Valverde, Mitzy Vado-Chacón, Diego Sossa-Rojas, Haipeng Liu, Dingchang Zheng

**Affiliations:** 1Industrial Design Engineering Department, Costa Rica Institute of Technology, Cartago 7050, Costa Rica; 2Electronics Engineering Department, Costa Rica Institute of Technology, Cartago 7050, Costa Rica; sarriola@itcr.ac.cr; 3Respiratory Therapy Department, Santa Paula University, San Jose 2633, Costa Ricadsossar@hotmail.com (D.S.-R.); 4Center of Intelligent Healthcare, Coventry University, Coventry CV1 5FB, UK

**Keywords:** pulse rate variability (PRV), heart rate variability (HRV), photoplethysmography (PPG), embedded systems, Bluetooth low energy (BLE), remote healthcare

## Abstract

Photoplethysmography (PPG) signals have been widely used in evaluating cardiovascular biomarkers, however, there is a lack of in-depth understanding of the remote usage of this technology and its viability for underdeveloped countries. This study aims to quantitatively evaluate the performance of a low-cost wireless PPG device in detecting ultra-short-term time-domain pulse rate variability (PRV) parameters in different postures and breathing patterns. A total of 30 healthy subjects were recruited. ECG and PPG signals were simultaneously recorded in 3 min using miniaturized wearable sensors. Four heart rate variability (HRV) and PRV parameters were extracted from ECG and PPG signals, respectively, and compared using analysis of variance (ANOVA) or Scheirer–Ray–Hare test with post hoc analysis. In addition, the data loss was calculated as the percentage of missing sampling points. Posture did not present statistical differences across the PRV parameters but a statistical difference between indicators was found. Strong variation was found for the RMSSD indicator in the standing posture. The sitting position in both breathing patterns demonstrated the lowest data loss (1.0 ± 0.6 and 1.0 ± 0.7) and the lowest percentage of different factors for all indicators. The usage of commercial PPG and BLE devices can allow the reliable extraction of the PPG signal and PRV indicators in real time.

## 1. Introduction

The photoplethysmography (PPG) signal reflects the volumetric changes in microcirculation. PPG signals can be measured by detecting light absorption at different body sites with dense microvasculature, such as the fingertips or the ear lobe [1,2]. Hemodynamic parameters can be extracted from the AC components of PPG signals for daily cardiovascular healthcare applications. The development of PPG-based remote healthcare infrastructure is creating a promising pathway toward next-generation healthcare by enabling a low-cost real-time healthcare framework, especially for low-resource communities in underdeveloped countries [3].

The PPG signal and its derivatives including velocity photoplethysmography (VPG) and acceleration photoplethysmography (APG) signals have been applied in investigating the physiological properties of the cardiovascular and respiratory systems including heart rate (HR), heart rate variability (HRV), and pulse rate variability (PRV) [4,5]. HRV indicators are often extracted from electrocardiogram (ECG) signals while the PRV indicators are extracted from the PPG signal. PRV indicators can be classified in accordance with the measurement time as long-term (≥24 h), short-term (~5 min), and ultra-short-term (≤5 min) ones. Ultra-short-term PRV monitoring can be achieved using the PPG signal. Peak-to-peak interval, defined as the distance between two systolic peaks in the PPG signal or its derivatives and is one of the commonest evaluations of PRV [6,7]. Other common PRV indicators include inter-beat interval (IBI), SD of normal-to-normal P-P intervals (SDNN), and root mean square of successive differences between normal heartbeats (RMSSD). However, the PPG signal and its derivatives can be affected by device properties, motion artifacts, and physiological factors such as anatomic variations, arterial stiffness, measurement site, posture, breathing pattern, and sentiment [8,9].

Several studies have examined the accuracy of PPG and APG signals in various scenarios, such as standing at rest, sitting with a deep breathing rhythm, and exercising [10]. Jan et al. investigated the coherence between ECG and PPG-derived PRV indicators during controlled and uncontrolled breathing patterns and found that PRV indicators were considerably less accurate during controlled breathing exercises [11]. Królak et al. studied the reliability of PRV extraction from prerecorded PPG data at rest and achieved a high approximation to ECG-derived HRV indicators [12]. PRV indicators from PPG and APG signals during rest and exercise have been extensively explored in controlled scenarios, with time-domain PRV indicators being accurately detectable at rest, while their accuracy during exercise varies across different studies [13,14,15,16,17]. Although comparison algorithms have been proposed to enhance the accuracy of time-domain PRV indicators, they have not been tested in real-time PPG signal extraction [10]. Moreover, there is currently a lack of comprehensive evaluation of different physiological factors and their effects on the accuracy of PPG-derived ultra-short-term PRV indicators, which are crucial in remote healthcare monitoring.

The integration of commercial pulse oximeters, or PPG sensors, with intelligent systems provides an affordable means for remote healthcare monitoring in low-resource settings, though their accuracy in many cases still needs validation [18]. Bluetooth low energy (BLE) is becoming increasingly popular in techniques for remote healthcare monitoring due to its low energy consumption and ability to transfer small data sets, especially in Internet of Things (IoT)-based applications. The use of PPG-based remote healthcare monitoring has the potential to revolutionize healthcare by reducing costs, enabling the agile implementation of healthcare infrastructure, providing universal healthcare access to entire populations, and decentralizing healthcare services [19,20].

Introducing new e-health technologies in Latin America holds great promise, as they have the potential to bring about numerous benefits. For example, IoT devices can enable healthcare providers to monitor the health status of patients in real time, facilitating early detection and treatment of potential health issues and ultimately improving overall patient outcomes. Additionally, remote monitoring can reduce the need for hospital readmissions, resulting in cost savings for both patients and healthcare providers. Furthermore, remote monitoring can empower patients to take an active role in managing their health, enhancing patient engagement [21].

When it comes to monitoring PRV remotely, previous studies have only examined PRV data in two or three different body postures. Additionally, although researchers have explored extracting PRV from pre-recorded remote PPG data, there has not been a thorough investigation into real-time PRV monitoring using affordable PPG systems [22,23].

To address the abovementioned gaps, the purpose of this paper is to investigate the standardized integration of commercial PPG and BLE devices for the extraction of HR, IBI, SDNN, and RMSSD at different body postures (sitting at 90°, sitting at 90° while deep breathing, standing, and supine).

## 2. Materials and Methods

### 2.1. Subjects

Thirty healthy male and female subjects (mean ± SD of age: 24 ± 5 years) from local communities without any known cardiovascular or respiratory disease participated in this study with written informed consent. The protocol was approved under resolution no. PE-OB-02-2021 of the Santa Paula University Ethical Committee. Before the experiment, the procedure was explained in detail to each participant.

### 2.2. Device Integration Architecture

The integration proposed in this study was proposed for low-resource centers that comply with the following premises:
Very low budget for medical equipmentTotal absence of WiFi equipmentPresence of a basic computer

Therefore, the integration (shown in Figure 1D) only includes the commercial PPG sensor, the BLE receiver unit connected to a basic computer that can locally store the data.

### 2.3. Equipment

The commercial Bluetooth Low Energy Pulse Oximeter (manufactured by Jumper Medical Equipment Co., Shenzhen, China, under the device model JPD-500G with a cost of USD 35) was used in this study as a non-invasive device that measures the oxygen saturation level in the blood through the PPG signal. It was equipped with an nRF52832 multiprotocol SoC chipset. The PPG sensor had a frequency range of 2.402–2.480 GHz and a power output of 0.4 mW, which adhered to the 15C radio frequency legislation. The sampling frequency of the PPG sensor was set to 40 Hz.

In contrast, the reference ECG sensor used in this study was the Polar H7 BLE Strap (manufactured by Polar Electro Oy, Helsinski, Finland), which has been validated in previous literature [24,25,26]. This sensor was placed on the participant’s chest to record the electrical activity of the heart. It had a sampling frequency of 1000 Hz, which provided a high temporal resolution and allowed for the capture of subtle changes in the heart’s electrical activity. The ECG sensor was wirelessly connected to the Kubios HRV software (ver. 1.1.10) through Bluetooth Low Energy technology, which ensured reliable and fast data transmission.

The receiving unit is composed of a 32-bit Extensa LX6 (manufactured by the M5Stack Company, Shenzhen, China, with a cost of 15 USD) dual nucleus chipset along with an in-built link transmitter and receiver of 2.4 GHz band compliant to Bluetooth v4.2/EDR and BLE specification. In addition, a laptop computer with 32 Gb of RAM and 1.5 TB of memory was used for data processing.

### 2.4. Measurement Procedure

Initially, the Polar H7 ECG strap was adjusted and positioned at the xiphoid process by the participant. Next, the heart rate BLE data transmission between the ECG sensor and the Kubios HRV app was confirmed for each participant. Following that, the commercial PPG sensor was placed on the participant’s right-hand index finger (as shown in Figure 1A), and the transmission of the PPG signal was validated through the serial port of the researcher’s personal computer. The HRV parameters were extracted using the commercial PPG sensor in four randomized scenarios: sitting at 90°, standing, and supine positions, all while breathing normally (as depicted in Figure 2), as well as sitting at 90° with deep breathing. Each test had a duration of three minutes [27] and was repeated twice with a 120 s break between the two trials. For the three postures under normal breathing, the participants were instructed to maintain stillness, avoid speaking, and breathe normally during each test. In the deep breathing scenario, the respiratory rate was controlled at approximately 10 breaths per minute. Throughout the experiment, a clinician was present to ensure that all the steps were correctly followed. The study protocol was designed to establish a foundation for physiological measurement in real-life situations. Therefore, we selected the three postures that are common in daily activities and have been widely implemented in prior studies, while the supine and deep breathing scenarios can simulate resting and sleeping states [28,29,30].

### 2.5. Pre-Processing of PPG Signals

As shown in Figure 3, to reduce the baseline wandering and signal saturation, a simple moving average filter was implemented on the real-time received raw data. Then, an adaptive iteratively reweighted penalized test [30] was implemented to stabilize the baseline of the PPG data. Consequently, a digital IIR sixth-order low-pass filter with a cutoff frequency of 18 Hz was applied. Filtering performance can be visualized in Figure 3, where noise is reduced significantly after the filter is applied (superimposed image). Once the filtering is implemented, the second derivative is applied to the PPG signal to utilize the APG component.

### 2.6. Extraction of Time-Domain HRV/PRV Indicators

#### 2.6.1. Extraction of HRV Indicators from ECG Signals

The ECG data were processed using Kubios HRV (ver. 1.1.10) software to extract HR, IBI, SDNN, and RMSSD [31].

#### 2.6.2. Extraction of PRV Indicators from PPG Signals

The precise detection of peaks was achieved using a Python-based algorithm developed from the Scipy Package (Figure 4) [32], which utilized a flexible decision-making rule that had been previously developed and compared in [33] to extract inter-beat intervals. In Formulas (1) and (2) the counter “k” for the cardiac events has been used. This counter runs from 1 to n, the total number of cardiac events to begin with. Cardiac event detection was initiated by identifying global maxima of the APG cardiac cycle, utilizing the outlined formula below to accentuate cardiac pulses, and assigning a value of zero to the remainder of the APG wave.
(1)$$z\iota =\sum_{k=i}^{n}\Delta \mu k$$
(2)$$\Delta \mu k=\left\{1:\Delta y(k)>ɵ,0:\Delta y(k)\leq ɵ\right\}$$

The logical array zι is created with i and n representing the first and nth elements, respectively, while ɵ denotes the acceptable threshold limit for peaks. N refers to the overall number of samples contained within the APG signal record, and Θ is the device’s sample frequency. The value of i is determined by the relative position within the data window ω, where ω=ω≤Θ/2<N. In the context of this study, the data window was established as 20 s.

$\Delta y\left(\kappa \right)=y\left(\kappa \right)-y\left(\kappa \pm \frac{\omega }{2}\right)$ is the local peak threshold.

Then, the extraction of peak-to-peak intervals was performed using the following formula:(3)PP=Δφ/Θ∗1000
where Δφ is the discrete distance from detected peaks (zι=1).

Then, heart rate was extracted as:(4)Mean Heart Rate=Measurement Time/mean(IBI)

SDNN was calculated as:(5)$$SDNN=\sqrt{\frac{1}{N-1}\sum_{n=2}^{N}{\left(RR(N)-RR\right)}^{2}}$$

RMSSD was derived as:(6)$$RMSSD=\sqrt{\frac{1}{N-2}\sum_{n=3}^{N}{\left(RR\left(N\right)-RR\left(N-1\right)\right)}^{2}}$$
where RR is the average peak-to-peak interval and N the number of cardiac events.

### 2.7. Statistical Methods

#### 2.7.1. Initial Analysis of HRV and PRV Indicators

All the statistical analyses were performed in the R programming language (Version 4.1.2) due to its extensive and powerful packages for data analysis and visualization such as ggplot2 and dplyr [34]. For descriptive statistics (mean ± standard deviation [SD]), Pearson correlation coefficient, percentage bias (P bias), and difference factors (DF%) [16] were calculated for all HRV and PRV indicators.

#### 2.7.2. Comparison between HRV and PRV Indicators in Four Physiological Status

In order to determine if PRV indicators can serve as reliable indicators for HRV, an assessment of normality was first conducted using the Shapiro–Wilk test. Normality was defined as having a *p*-value greater than 0.05 for all indicators in each of the four physiological statuses (i.e., three postures and deep breathing). Subsequently, a paired t-test was performed for each pair of HRV and PRV indicators that demonstrated normal distribution, while the Wilcoxon signed-rank test was utilized if normality was not met, in order to identify any significant differences between the two sets of indicators.

#### 2.7.3. Effect of Posture and Breathing Pattern on the Difference between HRV and PRV Indicators

To further explore how different postures may impact the difference between corresponding HRV and PRV indicators, an analysis of variance (ANOVA) or its non-parametric equivalent, the Scheirer–Ray–Hare test, was conducted. Prior to this, the homogeneity of variance was evaluated using Levene’s test, with homogeneity defined as having a *p*-value greater than 0.05. If homogeneity of variance was satisfied, the ANOVA was performed, whereas if it was violated, the Scheirer–Ray–Hare test was used instead. The independent variables for both tests included posture (sitting at 90°, standing, and supine), indicator type (HR, IBI, SDNN, and RMSSD), and their interaction. Post hoc analysis was then conducted using Tukey’s test for ANOVA, and Dunn’s test or Kruskal–Wallis test for each scenario in the Scheirer–Ray–Hare test, in order to identify the most reliable posture and HRV indicator. In a similar vein, the impact of breathing patterns on the relative difference between HRV and PRV indicators was examined through a comparison of results using either the paired t-test or the Wilcoxon signed-rank test for each type of indicator.

#### 2.7.4. Bland–Altman Analysis

To evaluate the accuracy of PRV indicators, the Bland–Altman analysis was performed between PRV and corresponding HRV indicators in all the postures and breathing patterns.

#### 2.7.5. Linear Regression Analysis

Correlation of PRV data between the PPG results and the ECG results was constructed to inspect if the data follow a linear pattern. Datasets from all the indicators were fitted to a linear regression model and the correlation was evaluated based on the regression coefficient (R^2^).

### 2.8. Data Loss Analysis

Average data loss was defined as the percentage of missing sampling points during the measurement. The mean and SD of data loss were calculated in each physiological status. In addition, the effect of posture on data loss was studied similarly to that of the difference between HRV and PRV indicators.

## 3. Results

### 3.1. Preprocessing and Filtering of the PPG Signal

Sampling frequency and pre-processing methods can significantly influence the signal quality of wearable sensors. In this study, a low-pass IIR filter was implemented to match the low sampling frequency of the device. For high-frequency noises, a next-step bandpass filter with frequency ranges between 3 to 5 Hz is recommended, although, it was not implemented to avoid aliasing [35].

### 3.2. Difference between HRV and PRV Indicators

In general, PRV indicators exhibited similar ranges (refer to Figure 5). A significant difference was observed between the RMSSD values derived from the PPG and ECG sensors while standing (*p* < 0.05 in paired t-test). Regarding the comparison of relative differences between HRV and PRV indicators, HR and IBI were found to have homogeneity of variance (*p* > 0.05 in Levene’s test) across the four physiological states, whereas SDNN and RMSSD did not exhibit homogeneity (*p* < 0.05). The physiological status did not show any significant effect on the results (*p* > 0.05); however, the type of indicator did (*p* < 0.05).

### 3.3. Post Hoc Analysis: Effect of Posture on the Difference between HRV and PRV Indicators

According to Dunn’s test for post hoc multiple comparisons analysis, among the three postures, the standing posture yielded the least reliable measurement, while sitting at 90° with a regular breathing pattern was found to be the most reliable one (*p* < 0.05 in Dunn’s tests). With respect to the indicators, the HR indicator showed a significant difference (*p* < 0.05 in Dunn’s test) in relative difference across all postures compared to other indicators, except for the IBI (*p* = 0.738, 1.000, and 0.932 in the sitting posture, supine and standing postures, respectively, in Kruskal Wallis tests). The distribution details of HRV and PRV are shown in Table 1, Table 2, Table 3 and Table 4. Although there was no significant influence of posture on the relative difference between PRV indicators overall, the most reliable posture was sitting at 90°, and the most reliable indicator was the HR, while the least reliable one was the RMSSD. Particularly, the PPG sensor tended to overestimate the measurement of RMSSD in the standing scenario, as shown in the P bias in Table 2.

### 3.4. Post Hoc Analysis: Effect of Breathing Pattern on the Difference between HRV and PRV Indicators

The normal and deep breathing patterns presented significant differences (*p* < 0.05) in HR and IBI with no significant difference in SDNN and RMSSD.

### 3.5. Bland–Altman Analysis between HRV and PRV Indicators

As shown in Figure 6, better dispersion was achieved for the SDNN indicator in the sitting posture while for the RMSSD indicator, it was in the sitting and standing postures. In comparison to [16] less dispersion was achieved for all the indicators in the sitting posture.

### 3.6. Linear Regression Analysis between HRV and PRV Indicators

The distributions of all indicators were aligned in all scenarios, as depicted in Figure 7. However, significant differences were observed for the SDNN indicator in the deep breathing scenario and for the RMSSD indicator in the standing posture. Strong correlations were found between HR and IBI for all postures. For the standing and supine postures, the SDNN and RMSSD showed weaker correlations and wider data dispersions, as shown in Figure 6 and Figure 7.

### 3.7. Data Loss Analysis

Table 5 displays the descriptive statistics for the frequency loss that was detected. The data loss was approximately 1% Hz across all postures. The Kruskal–Wallis test revealed that posture was a significant factor affecting data loss (*p* > 0.08), with the sitting posture during regular and deep breathing scenarios presenting the least significant difference. Thus, the sitting posture is recommended to minimize data loss.

## 4. Discussion

### 4.1. Reliability of PPG-Derived Ultra-Short-Term PRV Using the APG Component

Takazawa et al. have introduced the APG signal as a promising technique for estimating PRV. This method involves transforming the amplitudes of the canonical PPG signal to stress systolic phases of the cardiac cycles [36]. However, peak detection accuracy in the PPG signal can be distorted due to physiological abnormalities, whereas valley detection is not influenced by reflected waves and therefore presents higher accuracy. In the APG signal, valley detection is the zero-detection equivalent [35,37]. Outlier detection algorithms can also be used to identify the true APG signal features in practical applications, as altered breathing patterns can strongly influence the quality of data and create false positives [38].

Although long-term HRV recordings can predict health outcomes such as heart attacks or strokes, there is an increasing demand for ultra-short-term HRV monitoring due to its practicability in time-costly applications such as nursery control and routine medical practice scenarios. Furthermore, its potential for use in pain control has been explored. To assess ultra-short-term HRV features, various procedures have been proposed, but existing studies show inconsistency among the results [39]. Shaffer et al. [40] proposed an evaluation method that considers the normality of data, and other metrics include correlation analysis, Bland–Altman plots, and parametric or non-parametric statistical tests. This study follows a similar approach.

### 4.2. Accuracy of Time-Domain PRV Indicators in Different Postures and Breathing Patterns

The different factors of all the ultra-short HRV indicators (0.25%, 0.2%, 8.91%, and 11.1% for HR, IBI, SDNN, and RMSSD) were much lower than those in an existing study based on a commercial PPG device (1.61%, 37.99%, 9.96%, and 32.59% for HR, IBI, SDNN, and RMSSD) (Shown in Table 6) [16] at the sitting posture with regular breathing rhythm. Moreover, in the sitting posture, in comparison with [41] (Shown in Table 7), the short-term RMSSD indicators presented a considerable advantage (R^2^ = 0.82 vs. R^2^ = 0.56). On the other hand, the IBI, SDNN, and RMSSD presented better correlations in comparison to the Bora Band validated in [42] (Shown in Table 8). The deep breathing indicators performed slightly less in comparison to the state-of-the-art static BioSign system [43] (Shown in Table 9).

Body posture can have a significant impact on the accuracy and reliability of PPG signals, particularly in standing and motion postures where blood flow disruption and interference can occur. The sitting position has been found to provide prime accuracy and reliability, while the supine position presents acceptable recording capabilities. Additionally, the finger is a preferable measurement site for PPG signals as it allows for a higher overall filtering-induced time shift and lower intra-subject time shift variability compared to other sites [8,35].

Accurately estimating RMSSD and SDNN using PPG signals is a well-known challenge with high clinical significance, as RMSSD is an indicator for both atrial fibrillation (AF) and sudden unexplained death in epilepsy (SUDEP) [44], while SDNN measurement reflects sympathetic and parasympathetic nervous system activity [45]. Minor inaccuracies in IBI, which are common in PPG-derived data, can result in outliers and errors in RMSSD and SDNN. In contrast, ECG-derived HRV parameters are more stable due to the robustness of IBI calculated from R peaks against minor noises or filtering-induced waveform changes.

### 4.3. Accuracy Discussion of Time-Domain PRV Indicators in Different Postures and Breathing Patterns

Accurately detecting cardiac signals is a challenge in traditional HRV analysis, particularly in low respiratory rates where activity shifts into lower frequencies of parasympathetic activity can overlap with the sympathetic region, leading to reduced accuracy. While ECG remains the standard for capturing HRV indicators in altered breathing patterns by providing isolation of sympathetic and parasympathetic activity [46], the approach implemented in this study provides an accurate and reliable reference for time-domain PRV indicators at different respiration rates using significantly less expensive equipment compared to state-of-the-art systems.

The study found that better results were achieved in the sitting posture at 90° degrees at rest, and acceptable accuracy was achieved in the stand scenarios for comparable indicators. However, no similar study has been conducted for the supine position, and posture alteration can cause changes in the quality of PPG signals and shift PPG feature points, leading to errors in HRV measurement [8]. Therefore, the validation of wearable sensors in different postures, particularly in the supine position, is a current challenge that was addressed in this study following the procedure stated in [27].

The results showed that using commercial PPG and BLE devices for real-time extraction of time-domain HRV indicators have acceptable accuracy, high scalability, stability, and minimum data loss. The deviation was observed in the standing posture, and the accuracy was better than many sensors reported in the literature, specifically in the accuracy of the RMSSD indicator [16].

### 4.4. Application of Commercial PPG Sensor and BLE Receiver: Towards Standardization

BLE, which was the standard communication protocol implemented in this study, presents several advantages for short-distance applications with low latency, low implementation complexity, and low energy consumption with high availability that allows community support for the development of the technology. However, it only enables short-distance applications, which is a major limitation compared with WiFi and 5G technologies [47]. To address this gap, some researchers are exploring approaches to diminish the packet loss rate at dense BLE device areas with different distances [48] where the average range is up to 100 m while the latency is considerably less than the classic Bluetooth protocol, although, range and power efficiency depend on environmental conditions, antenna strength, and presence of obstacles [49].

The wireless configuration between the PPG sensor and the embedded unit followed the controller–host stack principle [50] which operates in the 2.4 GHz ISM band (industrial, scientific, and medical) and states forty radio-frequency channels in two categories, i.e., advertising channels and data channels [51]. This property has the advantage of a frequency hopping mechanism established at the top of the data channels configured via the ATT-GATT byte profiles, a feature that diminishes radio interference and wireless propagation issues [52].

In addition to its acceptable accuracy, the system integration implemented in an extraction–transfer–load approach in this study was a low-cost solution (shown in Figure 8A) that can be standardized and implemented in real-world remote healthcare monitoring (shown in Figure 8B), which provides an affordable solution to improve the healthcare in low-resource settings. The standardization of real-time transmission, cloud-based real-time signal processing, and in-depth data analysis can promote the large-scale commercial application of PPG-based healthcare services with affordable equipment. A cloud-based infrastructure that can leverage the signal processing framework using a highly available and reliable system is proposed in Figure 8. Using scalable and remotely maintainable infrastructure can leverage data access for clinicians, patients, researchers, data scientists, and IT experts. Cloud services provide several facilities such as constant technical support, shared-responsibility maintenance principles, and end-to-end encryption of patient data. These facts allow clinicians to focus only on clinical updates without sacrificing service coverage and reliability [53,54]. Currently, commercial oximeters and PPG devices do not possess integration patterns and approaches for the implementation in e-health routine monitoring applications, especially in low-resource communities, which is the gap we aimed to fill.

### 4.5. Remote Healthcare Monitoring in Rural Areas: Latin America and the World

Latin America is a region with limited progress in remote healthcare due to the lack of relevant regulations, underdevelopment of the biomedical engineering industry, low public awareness of remote healthcare, as well as other political, social, and infrastructural factors. However, there is a high need for remote healthcare in Latin America considering the scarceness of medical resources in rural areas, low population density, and social inequality between highly urbanized and rural areas [22]. Beyond Latin America, low-cost, publicly accessible remote healthcare is a pressing necessity for low-resource areas globally. This integral remote monitoring philosophy was proposed by Goodbridge et al. to further comprehend the outcomes of cost-effective systems in case-oriented environments since approximately 50% of the global population lives in rural or similar communities [55,56]. The protocol of this study was designed to lay the groundwork for future physiological research in real-world settings where the three postures are common in daily activities, while the deep breathing and supine scenarios can simulate the resting and sleeping states [29,57].

### 4.6. Limitations and Future Directions

The sampling frequency of the commercial device was the major limitation of this study. The high-frequency sampling can capture more details of cardiac behavior and enable the detection of frequency-domain HRV/PRV indicators such as high-frequency (HF) and low-frequency (LF) components and their ratio. Frequency samplings over 500 Hz are recommended for a full spectra analysis [10]. In this pilot study, we only tested three typical postures without dynamic monitoring. In addition, we included a limited number of subjects who were all healthy. In future studies, high-performance wearable sensors such as the Polar H10 model can be used to explore the extraction of frequency-domain HRV/PRV parameters, more postures, and dynamic monitoring in real-world application scenarios [58]. The integrations with multifunctional sensors such as handheld or wristband devices and the inclusion of chronically ill patients are also recommended for future research.

## 5. Conclusions

Physiological abnormalities and body posture, particularly sitting, can affect the accuracy of PPG signals. However, there is a lack of research on the impact of the supine position. This study demonstrated that using commercial PPG and BLE devices with real-time extraction of time-domain HRV indicators is achievable with acceptable accuracy, scalability, stability, and minimal data loss. The study introduced an evaluation method that considers data normality, correlation analysis, Bland–Altman plots, and parametric or non-parametric statistical tests. The standardized approach provides an accurate and dependable reference for time-domain PRV indicators at different respiratory rates, utilizing cost-effective equipment compared to high-performing commercial systems. Although all postures showed acceptable accuracy, it is not recommended to use motion-induced scenarios for the real-time extraction of HRV and PPG-related indicators. The solution is scalable, low-cost, and integrable.

The proposed approach is not only accurate and reliable but also has the advantage of being less expensive compared to compared systems. This makes it an ideal solution for low-income countries where healthcare resources may be limited. Additionally, the proposed evaluation method provides a standardized and objective way to assess the quality of the data, which can be particularly useful in resource-limited settings where expert interpretation of data may not be readily available. Overall, this approach has the potential to facilitate the widespread use of PRV analysis as a non-invasive and low-cost tool for cardiovascular health monitoring in low-income countries.

## Figures and Tables

**Figure 1 sensors-23-04246-f001:**
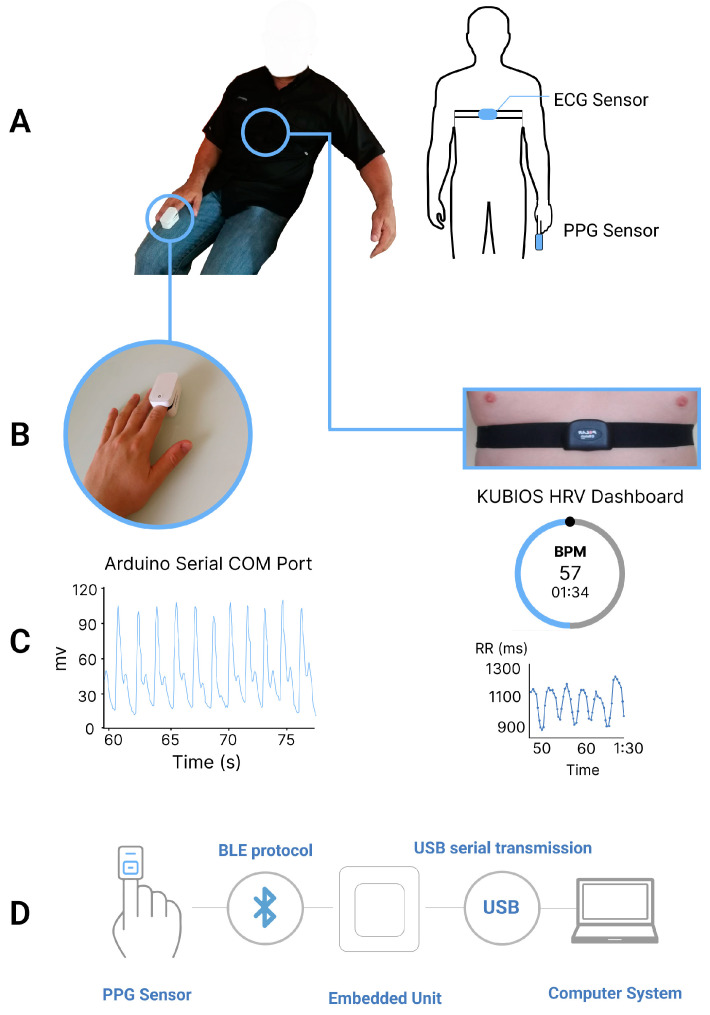
Technical sensor setup (**A**) sensor placement (**B**) enlarged view of the sensors, (**C**) PPG signal extraction and ECG HRV parameter processing with the Kubios HRV App (**D**) work diagram of the entire system.

**Figure 2 sensors-23-04246-f002:**
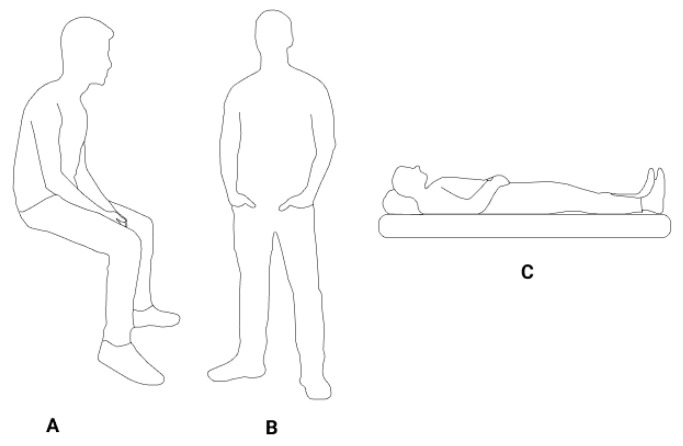
Postures in the test (**A**) sitting at 90°; (**B**) standing; (**C**) supine. The subjects breathed regularly in standing and supine postures. In sitting posture, the measurement was repeated in regular and deep breathing patterns for comparison.

**Figure 3 sensors-23-04246-f003:**
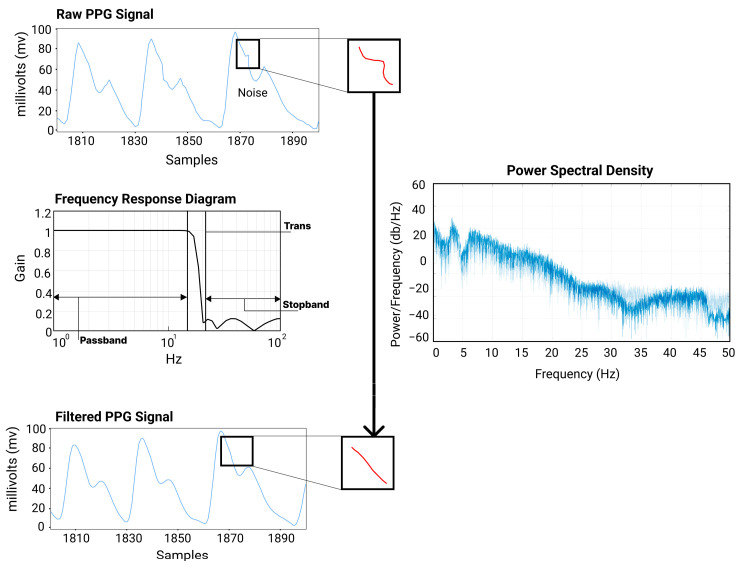
Filtering method diagram of the raw PPG signal. Noise augmentation is shown in red curves.

**Figure 4 sensors-23-04246-f004:**
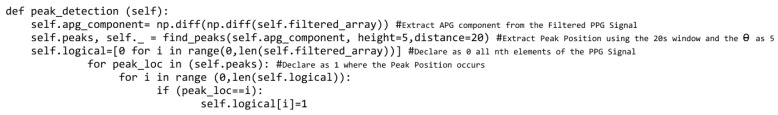
Python algorithm implemented.

**Figure 5 sensors-23-04246-f005:**
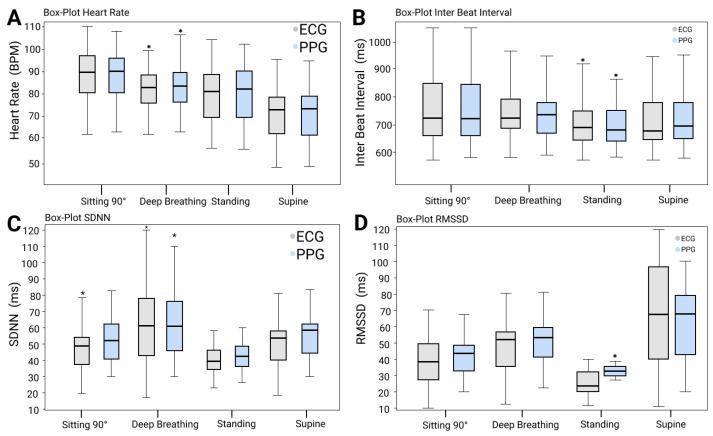
Boxplot and median analysis for all HRV and PRV indicators derived by ECG and the commercial PPG sensor in the four different positions. Low and high box borders represent the first and third quartiles, respectively. * Symbol represents data outliers.

**Figure 6 sensors-23-04246-f006:**
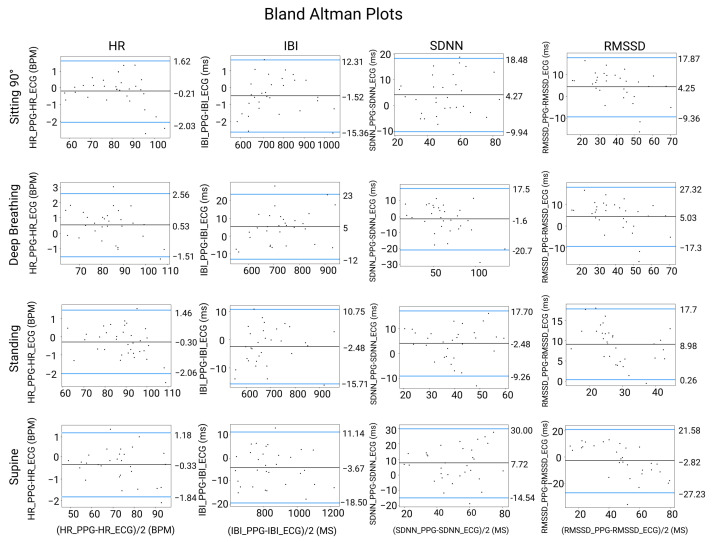
Bland–Altman analysis between PRV and HRV indicators in four physiological statuses, where _PPG and _ECG denote PPG- and ECG-derived results, i.e., PRV and HRV indicators. The x axis represents the average mean values between ECG- and PPG-derived measurements and mean values of the PPG-derived measurement while y axis represents the average difference between ECG and PPG derived measurements.

**Figure 7 sensors-23-04246-f007:**
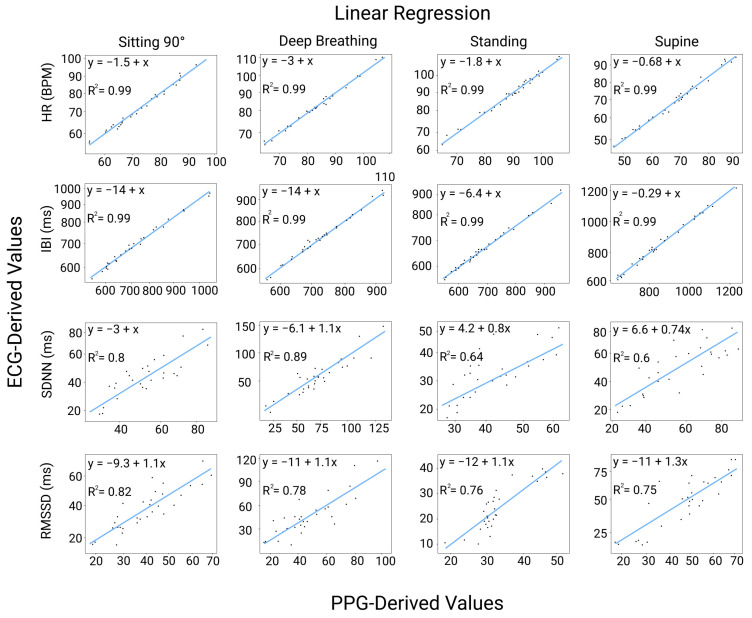
Linear regression analysis between HRV and PRV indicators in four physiological statuses.

**Figure 8 sensors-23-04246-f008:**
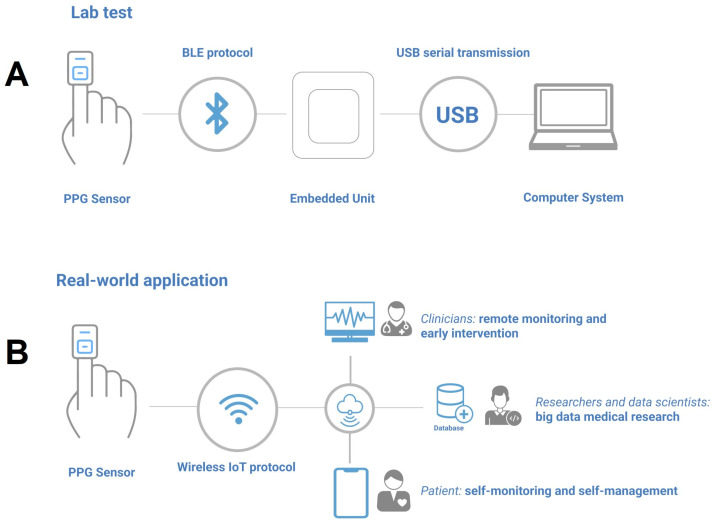
PPG-based remote healthcare monitoring. (**A**) Settings for lab tests utilized in this study. The BLE protocol configured using the ATT-GATT standard was implemented between the PPG sensor and the embedded unit. The embedded unit was connected using a standard 3.0 USB protocol. (**B**) A cloud-based standardized solution for real-world applications for low-resource communities. The sensors send the raw PPG data through IoT for cloud-based processing.

**Table 1 sensors-23-04246-t001:** Descriptive statistics of HRV performance in the sitting at 90° posture. HRV: heart rate variability; PRV: pulse rate variability; ECG: electrocardiogram; PPG: photoplethysmography; SD: standard deviation; RMSE: root mean square deviation.

PRV Indicator	ECG-Derived HRV (Mean ± SD)	PPG-Derived PRV (Mean ± SD)	ECG-PPG Difference (Mean ± SD)	RMSE	P Bias	DF %
Heart rate (BPM)	81.44 ± 12.98	81.23 ± 12.68	0.21 ± 0.3	0.94	0.3	0.26
NN interval (ms)	757.16 ± 130.38	758.69 ± 130.32	1.53 ± 0.06	7.11	−0.2	0.2
SDNN (ms)	47.94 ± 14.97	52.21 ± 15.96	4.27 ± 1	8.31	−8.2	8.91
RMSSD (ms)	38.33 ± 15.91	42.58 ± 12.88	4.25 ± 3.03	8.05	−10	11.11

**Table 2 sensors-23-04246-t002:** Descriptive statistics of HRV performance in the standing posture. HRV: heart rate variability; PRV: pulse rate variability; ECG: electrocardiogram; PPG: photoplethysmography; SD: standard deviation; RMSE: root mean square deviation.

PRV Indicator	ECG-Derived HRV (Mean ± SD)	PPG-Derived PRV (Mean ± SD)	ECG-PPG Difference (Mean ± SD)	RMSE	P Bias	DF %
Heart rate (BPM)	88.58 ± 11.74	88.29 ± 11.44	0.29 ± 0.3	0.93	0.3	0.34
NN interval (ms)	733.66 ± 101.19	728.62 ± 100.39	5.04 ± 0.8	10.32	0.7	0.69
SDNN (ms)	63.15 ± 10.71	61.57 ± 10.68	1.58 ± 0.3	9.7	2.6	2.5
RMSSD (ms)	24.39 ± 9.01	33.37 ± 7.21	8.98 ± 1.8	9.99	26.9	36.82

**Table 3 sensors-23-04246-t003:** Descriptive statistics of HRV performance in the sitting at 90° posture while deep breathing. HRV: heart rate variability; PRV: pulse rate variability; ECG: electrocardiogram; PPG: photoplethysmography; SD: standard deviation; RMSE: root mean square deviation.

PRV Indicator	ECG-Derived HRV (Mean ± SD)	PPG-Derived PRV (Mean ± SD)	ECG-PPG Difference (Mean ± SD)	RMSE	P Bias	DF %
Heart rate (BPM)	83.28 ± 11.52	83.81 ± 11.15	0.53 ± 0.37	1.15	−0.6	0.63
NN interval (ms)	733.66 ± 101.28	728.62 ± 98.39	5.04 ± 2.89	10.32	0.7	0.69
SDNN (ms)	63.15 ± 27.96	61.57 ± 23.46	1.58 ± 4.5	9.7	2.6	2.5
RMSSD (ms)	46.55 ± 23.67	51.58 ± 18.58	0.53 ± 0.40	12.26	−9.7	10.8

**Table 4 sensors-23-04246-t004:** Descriptive statistics of HRV performance in the supine posture. HRV: heart rate variability; PRV: pulse rate variability; ECG: electrocardiogram; PPG: photoplethysmography; SD: standard deviation; RMSE: root mean square deviation.

HRV/PRV Indicator	ECG-Derived HRV (Mean ± SD)	PPG-Derived PRV (Mean ± SD)	ECG-PPG Difference (Mean ± SD)	RMSE	P Bias	DF %
Heart rate (BPM)	72.71 ± 12.86	72.38 ± 12.66	0.33 ± 0.2	0.82	0.5	0.45
NN interval (ms)	851.51 ± 155.04	855.18 ± 155.48	3.67 ± 0.44	8.29	−0.4	0.43
SDNN (ms)	48 ± 16.48	55.72 ± 17.16	7.72 ± 0.68	13.58	−13.9	16.1
RMSSD (ms)	49.52 ± 22.94	46.69 ± 15.19	2.83 ± 7.75	12.57	6	5.7

**Table 5 sensors-23-04246-t005:** Data loss in different physiological statuses.

Scenario	Data Loss (%)
Sitting at 90°	1.0 ± 0.6
Sitting at 90° in deep breathing	1.0 ± 0.7
Standing	1.0 ± 0.3
Supine	1.0 ± 0.7

**Table 6 sensors-23-04246-t006:** PRV accuracy comparison with the Empatica E4 validated in [16] for the sitting posture while breathing at regular intervals.

PRV Indicator	Jumper Commercial BLE Oximeter %DF	Empatica E4 Validated in [16] %DF
Heart rate (BPM)	0.25%	1.61%
NN interval (ms)	0.2%	37.99%
SDNN (ms)	8.91%	9.96%
RMSSD (ms)	11.1%	32.59%

**Table 7 sensors-23-04246-t007:** PRV accuracy comparison with the Flowmet LAS device validated in [41] for the sitting posture while breathing at regular intervals.

PRV Indicator	Jumper Commercial BLE Oximeter R^2^	Flowmet LAS Device Validated in [41] R^2^
SDNN (ms)	0.8	0.82
RMSSD (ms)	0.82	0.56

**Table 8 sensors-23-04246-t008:** PRV accuracy comparison with the Bora Band device validated in [42] for the sitting posture while breathing at regular intervals.

PRV Indicator	Jumper Commercial BLE Oximeter R^2^	Bora Band Device Validated in [42] R^2^
NN interval (ms)	0.99	>0.7
SDNN (ms)	0.8	>0.7
RMSSD (ms)	0.82	>0.7

**Table 9 sensors-23-04246-t009:** PRV accuracy comparison with the BioSign HRV system validated in [43] for the sitting posture while deep breathing.

PRV Indicator	Jumper Commercial BLE Oximeter R^2^	BioSign HRV System Validated in [43] R^2^
HR (bpm)	0.99	1.00
SDNN (ms)	0.8	0.997
RMSSD (ms)	0.82	0.982

## Data Availability

Data sharing is unavailable due to compliance with the informed consent provided to all subjects.

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
