# Peer review of "Real-Time Evaluation of Time-Domain Pulse Rate Variability Parameters in Different Postures and Breathing Patterns Using Wireless Photoplethysmography Sensor: Towards Remote Healthcare in Low-Resource Communities"

_sensors, 2023, doi:10.3390/s23094246_

Round 1
Reviewer 1 Report
The topic of this paper, dealing with “Parameters in Different Postures and Breathing Patterns Using Wireless Photoplethysmography Sensor: Towards Remote Healthcare in Low-Resource Communities”, seems to be interesting.
However, the paper needs significant improvement in the following areas:
1. The paper is verbose in nature. Many sections of the text explain general ML-DL principles, which is completely superfluous for this level of technical article.
2. The authors did not properly specify the underlying assumptions behind the suggested architecture. In general, it is not possible for a realistic system to function under every potential circumstance.
3. The features of the proposed system should be specified in detail.
4. It is unclear why the authors switched from Python to R for statistical analysis, given that Python is regarded as more effective for such tasks.
5. “The sitting posture in both breathing patterns showed the least data loss”—in Abstract need to be quantified.
6. The motivation of the work is not clearly explained.
7. Few figures, such as Fig. 3, have labels that are hardly distinguishable.
8. How could the authors ensure that the proposed framework is superior to others? It is suggested to show the performance evaluation table and graphs comparing the proposed framework to the related studies.
9. The system's accuracy calculation is completely absent.
10. It is suggested that a flowchart or working diagram of the entire system be included.
11. There was no reference from any work of 2022. It is suggested to update the references and compare the present work with few of them.
Author Response
Dear Reviewer,
Thanks for your improvement feedback.
- We have removed superfluous term and have re written the majority of the content in order to provide a better understanding.
- We have added under section 2.1 the assumptions behind the architecture proposed.
- We have added all the features of the proposed system, including brands, model number and average price.
- We have added the justification of why we used R for statistical analysis.
- Quantification has been added to the abstract.
- Motivation of the work has been added at the introduction. (IoT remote healthcare devices is still un explored in latin america.
- Figures have been revised and improved where all the labels are now readable.
- We have added 3 tables for accuracy comparison with commercial high performing devices such as the Empatica E4, Flowmet LAS, Bora Band and the BioSign static HRV system (clinical device). It is important to note that newest devices do not mean more accurate ones. Our objective is to compare with commercial equipment that performs well under several conditions and postures and are used by not only academia but as well for clinical centers.
- Accuracy has been improved by adding comparison to more systems and as well the section has been re written for a more complete understanding.
- A work diagram has been added to Figure 1B.
- We have updated the references accordingly to more newer versions. For the device comparison we have added a study that involves a commercial validated system on 2022 with ULTRA-Short PRV Taoum, A.; Bisiaux, A.; Tilquin, F.; Le-Guillou, Y.; Carrault, G. Validity of Ultra-Short-Term HRV Analysis Using PPG—a Preliminary Study. Sensors 22 no. 20. 2022, 7995. https://doi.org/10.3390/s22207995.
Many Thanks,
Felipe Pineda

Reviewer 2 Report
INTRO: There is a clear need to quantitatively evaluate the value of different HRV/PRV indicators for different posture and health conditions for low-cost PPG solutions for remote healthcare applications in LMI countries. The presented analysis methods are appropriate. However, general remarks to improve the quality of the paper are as follows:
- Better and consistent usage of PRV and HRV. They appear too often mixed in the text. In the introduction the difference should be once clarified and then not anymore mixed.
- Often a lack of clear logic in the formulations of conclusions at the end of different sections in the text
- Formulas: some formulas contain mistakes and some clarifications needed
A major revision is needed to improve substantially the fluent reading of the text. Often the reader has to reread several sections, because the logic in conclusions is not clear. Some formulas contain mistakes and for some formulas there is a need for better clarifications.
More specific requests/remarks:
1/ Introduction
- Parameters and indicators are mixed in the story.
- PRV and HRV are intermixed in the story.
- Please mention once clearly in the text their definitions and differences
o HRV is related to ECG
o PRV is related to PPG
and then avoid the mixed usage
2/ Which commercial PPG system has been exactly used?
3/ Table descriptors (Table 1, 2, 3, 4): should contain HRV and PRV
4/ Fig. 3 Would it be possible to superimpose the power spectral density functions and discuss their difference?
5/ Fig. 5. The x axis represents the average mean values.
6/ Please clarify better the formulas (1) and (2): what about “Sum Slope: function?
7/ Formula (4): ???? ????? ???? = ??????e???? ????/????(???)
8/ Formula (5): RRj = please use subscript and RR = averaged RR?
9/ Formula (6): RMDSS = not correct!
10/ Section 3.1 – Section 3.2 – Section 3.5 – section 3.6
- Please formulate more clear conclusions.
11/ Formatting of the tables 1,2,3,4 can be more adapted to the page width
12/ Section 4.1
This section discusses the preprocessing. It would make sense to discuss this before the presentation of the generated data.
About APG: It is not clear if this derived signal has been used or not. It should be earlier in the text. That part plus the second part of this section belong rather to the introduction section.
13/ 4.4. Application of commercial PPG sensor and BLE receiver: towards standardization
- Which distances have been used? Can you indicate some ranges of distances? Short , long between different systems (BLE), Wifi, ...
14/ Section: limitations and future directions
The high-frequency sampling can capture more details of cardiac behavior and enable the detection of frequency-domain HRV-PRV indicators. Please indicate how much more HF-sampling is needed?
15/ Unified format in the reference list is needed. Vol/no, page numbers are most of the time missing.
Author Response
Dear reviewer,
Thanks for your comments and more complete revision.
As per your observations:
- We have re written the introduction and removed duplicated usage of HRV and PRV accordingly. We have improved our grammar in order to avoid confusion as well.
- We have added all the details in regards to the commercial BLE PPG device used including brand, model and price.
- Unfortunately this comment does not provide any meaningful insight and we were not able to deduct what you meant. We have improved the font size and image quality for all our images.
- We have superimposed the images and comment accordingly.
- 6. 7. 8. 9. We have updated all the formulas for SDNN and RMSSD.
- We have provided a more wide explanation for the cardiac event detection formulas + adding the Python code snippet. This will bring more clarity of the how-to replicate the logic and will bring more insights on the behind the scenes technical workflow.
10. We have re written all our discussion and conclusions
11. Table reformat can be adjusted with the MDPI team. The standard document provides a lot of limitations from a layout perspective. We have improved table size as much as possible.
12. We have bring the pre processing discussion before presenting the data and have clarified that we use the APG signal once the filtering is implemented. We have added our code snippet in python in order to provide a 360 view to our readers on how to implement the formulas for cardiac detection.
13. We have added the distance range for BLE in comparison to the bluetooth stack with the appropriate reference.
14. We have included what is the frequency required to capture the majority of cardiac behavior from a PPG perspective with the appropriate reference.
15. We have updated and reformat all of our references with a standard format that includes vol, no and doi.
Overall your revision has been more specific and applicable to apply improvements. We appreciate that.
All the best,
Felipe Pineda

Round 2
Reviewer 2 Report
Dear authors,
thank you for the in-depth response to the reviewer's input. The quality and the readability of the paper have been seriously upgraded. Also the reference list looks now professional.
My final remarks are related to the formulas
In formulas (1) and (2) the counter "k" for the cardiac events has been used. This counter runs from 1 till "n:, the total number of cardiac events.
- Don't switch to "kappa" in the formula on " line 212" just below these formulas (1) and (2).
- Probably better to switch to "N" instead of "n" as in the rest of the formulas, you use "N".
In formula (5), RR is not defined. It should be defined as the average as indicated in my first review.
Formula (6): a closing bracket is missing just before the exponent "2".
Best regards from the reviewer.
Author Response
Dear reviewer,
Many thanks for your revision.
I am glad there were minor revisions.
Kindly find attach the article with your improvements. I have added this sentence due to its great value to wrap up the goal of the formula we have developed:
-In formulas (1) and (2) the counter "k" for the cardiac events has been used. This counter runs from 1 till "n:, the total number of cardiac events.
Best regards,
Felipe